# Endoscopic Ultrasound-Guided Treatments for Non-Variceal Upper GI Bleeding: A Review of the Literature

**DOI:** 10.3390/jcm9030866

**Published:** 2020-03-21

**Authors:** Claudio Giovanni De Angelis, Pablo Cortegoso Valdivia, Stefano Rizza, Ludovica Venezia, Felice Rizzi, Marcantonio Gesualdo, Giorgio Maria Saracco, Rinaldo Pellicano

**Affiliations:** Gastroenterology and Digestive Endoscopy Unit, AOU Città della Salute e della Scienza, University of Turin, 10126 Turin, Italy; eusdeang@hotmail.com (C.G.D.A.); cortegosopablo@yahoo.it (P.C.V.); ludovica.venezia@gmail.com (L.V.); rizzifelice91@libero.it (F.R.); marcantonio.gesualdo@gmail.com (M.G.); giorgiomaria.saracco@unito.it (G.M.S.); rinaldo_pellican@hotmail.com (R.P.)

**Keywords:** EUS, bleeding, hemostasis, Dieulafoy, pseudoaneurysm, endotherapy

## Abstract

Endoscopic injection of glues, clotting factors, or sclerosing agents is a well-known therapy for the treatment of non-variceal upper gastrointestinal bleeding (NVUGIB), but less is known about endoscopic ultrasound (EUS)-guided treatments. In this setting, literature data are scarce, and no randomized controlled trials are available. We performed a review of the existing literature in order to evaluate the role of EUS-guided therapies in the management of NVUGIB. The most common treated lesions were Dieulafoy’s lesions, pancreatic pseudoaneurysms, and gastrointestinal stromal tumors (GISTs). Mostly, the treatments were performed as a salvage option after failure of conventional endoscopic hemostatic attempts, showing good efficacy and a good safety profile, also documented by Doppler monitoring of treated lesions. EUS-guided therapies may be an effective option in the treatment of refractory NVUGIB, thus avoiding radiological or surgical management. Nevertheless, available literature still lacks robust data.

## 1. Introduction

Non-variceal upper gastrointestinal bleeding (NVUGIB) is the most common type of acute upper gastrointestinal (GI) bleeding, accounting for 50–160 patients per 100,000 per year [1]. The mortality rate caused by NVUGIB remains high despite medical and technical improvements, probably due to longer life expectancy of the population [2], even though a decrease has been recorded in nationwide databases in the last 20 years [3]. Peptic ulcer is the single most common cause of NVUGIB, involving 25–67% of cases [4]; other common causes include erosive disease, Mallory–Weiss syndrome, Dieulafoy’s lesions, gastric antral vascular ectasia (GAVE), or less frequent vascular lesions such as peripancreatic pseudoaneurysm.

In recent years, endoscopic tools for achieving hemostasis have evolved, with a progressive increase of the success rate of therapeutic maneuvers. Novel technologies have been developed (i.e., new rotatable clips, Over-the-scope-clips (OTSC) clipping system, Padlock^TM^ clips), and new hemostatic agents have been introduced in clinical practice (i.e., hemostatic powders or a novel self-assembling matrix-forming gel) [5,6,7,8,9,10].

Nevertheless, despite the evolution of endoscopic hemostatic techniques and tools, some patients are refractory to the standard therapy and need different approaches, such as the surgical or radiological ones. In this setting, endoscopic ultrasound (EUS)-guided techniques could offer, at least theoretically, a feasible option in the treatment of patients with NVUGIB who are refractory to or not eligible for standard therapies.

Therapeutic EUS has become the first option for the treatment of many pathological conditions in recent years due to its widespread diffusion and the upcoming availability of new accessories. The availability of minimally invasive treatments combined with the vascular access from the lumen of the GI system under EUS guidance have made EUS an interesting potential alternative for the treatment of these hemorrhagic conditions.

EUS offers access to many intra-abdominal and mediastinal vascular structures that are in close proximity with the GI tract and easy access to the vessels that are located in the gut wall via needles which can be clearly recognized in the ultrasonographic field producing a thin hyperechoic path with an acoustic shadow behind it [11]. Until recent times, these vessels have only been accessible to surgeons and interventional radiologists: EUS guidance now offers an attractive, minimally invasive, alternative access route for therapeutic vascular intervention. So far, EUS-guided vascular therapies have found greater application in the field of portal hypertensive GI bleeding, but also in the field of NVUGIB: techniques such as injections of glue, clotting factors, or sclerosing agents and insertion of embolization coils have been described [12,13]. Furthermore, an interesting application of endoscopic ultrasound in the setting of NVUGIB could be the Doppler endoscopic probe (DEP) which may be used at the time of diagnosis and treatment of an acute episode, in the risk stratification for rebleeding and clinical outcome [14,15]. Although the preliminary results of using this tool are promising, the technique is complex, and results are hardly reproducible. For this reason, current guidelines agree that it is still too early to make evidence-based recommendations on its use [16,17].

Cyanoacrylates are a class of synthetic glues that rapidly solidify on contact with weak bases, such as water and blood, produced by several manufacturers. Cyanoacrylate glues can be mixed with Lipiodol: the rate of solidification is slower, but the endoscopic administration via needle injection is facilitated, and the risk of inadvertent adherence to catheters and endoscopes is reduced [18]. Cyanoacrylate glues have a well-established role in the endoscopic management of GI variceal bleeding. Despite their relative safety, there are concerns about potential severe complications, mainly, distant embolization; indeed, cyanoacrylate is considered a rescue therapy for achieving endoscopic hemostasis in high surgical risk patients [19,20]. EUS-guided injection of glues is performed via a 19 to 21 gauge (G) needle in or around an actively bleeding point or a non-bleeding vessel.

The fibrin sealant contains two components: human fibrinogen with clotting factor XIII and a starter solution containing human thrombin, reconstituted in 2 separate syringes before use. When mixed, these agents form a clot by mimicking the terminal phase of the physiological clotting cascade producing a strong cross-linked fibrin polymer. Fibrin glue is commercially available from several sources, and it is used in Europe for endoscopic hemostasis in bleeding ulcers and varices [18].

Among all available sclerosing agents, absolute alcohol is the least expensive, but when used for the hemostasis of esophageal varices, it has been associated with deep esophageal ulcers and tight stenosis with a high risk of rebleeding. In one study, alcohol was injected through a 22 G fine needle aspiration (FNA) needle for the treatment of a pancreatic pseudoaneurysm and a Dieulafoy’s lesion [21].

Sclerotherapy using polidocanol 1% or a combination of polidocanol 1% and epinephrine (1:10,000) with a 23 G needle was described in a study by Fockens and colleagues [22].

Coils can be placed alone or in combination with cyanoacrylate glues; thus, the coils could serve as a scaffold to which the glue could anchor in order to reduce the risk of glue embolization [23].

To date, EUS-guided therapies in NVUGIB have been performed both in naïve patients (as the first intervention) and in patients refractory to standard treatments. In this context, Dieulafoy’s lesions and pancreatic pseudoaneurysms are the best described lesions, since other lesions, such as peptic ulcers, are generally more eligible for and responsive to standard hemostatic techniques.

Nevertheless, at present, the role of EUS in this field still lacks large data supporting its use compared to standard therapies. After the first paper published by Fockens and colleagues in 1996, the literature on this topic remained scarce and limited to case reports and case series [22].

The aim of our paper was to review all available literature data about EUS-guided therapies in patients suffering from NVUGIB, evaluating the feasibility, the effectiveness, and the possible complications of these techniques.

## 2. Materials and Methods

### Study Selection

We analyzed Medline/Pubmed, Scopus and Web of Science databases in order to identify articles investigating the role and the applications of EUS-guided therapy in the treatment of NVUGIB. The keywords used were “EUS-guided therapy”, “non-variceal upper GI bleeding”, “peptic ulcer”, “Dieulafoy lesion”, “esophageal neoplastic lesions”, “gastric neoplastic lesions”, “duodenal neoplastic lesions”, “melena”, “pancreatic pseudoaneurysm”, and “sclerotherapy”.

We included all the studies reporting data on the role of EUS-guided therapy in the hemostasis of NVUGIB excluding those not explicitly taking into consideration the lesions different from varices and those in which lesions were not actively bleeding or in which the treatment was prophylactic [23,24,25].

The main outcomes considered in our review were technical and clinical success, complications of the procedures, and immediate and delayed adverse events. Previous treatments (mainly endoscopic or angiographic), type of lesion, cause and type of bleeding, type of needle, and type of treatment were recorded.

Due to the peculiarity of the topic and the paucity of the studies reported in literature, only case series and case reports were found and included. There were no restrictions on the publication date of the studies, and the final date of the search was August 30, 2019. For completeness and transparency, the quality level of the included studies was assessed using the consensus-based clinical case reporting guideline (CARE) [26].

## 3. Results

### 3.1. Study Characteristics

The literature search resulted in 120 publications overall. After screening all titles and abstracts for suitability, a total of 12 publications (4 case series and 8 case reports) were considered eligible to be included in the review.

The papers including a total of 41 patients were collected; out of these, 35 patients were treated with the EUS-guided therapy for NVUGIB (20 males and 12 females; in 3 cases, gender was not specified) and were eventually included (Table 1); 6 patients were excluded because EUS-guided treatments were performed on either variceal or colorectal lesions (2 and 4 cases, respectively). Overall, 39 EUS-guided procedures were performed.

The mean age at the time of the procedure was 62 years (range 17–94). These studies were published in the period between 1996 and 2019 [21,22,27,28,29,30,31,32,33,34,35,36].

The following causes of NVUGIB were reported: Dieulafoy’s lesions (10), pancreatic/vascular pseudoaneurysms (9), GI stromal tumors (GISTs) (7), intractable marginal ulcer after a Roux-en-Y gastric bypass (1), duodenal metastasis from colon cancer (1), duodenal metastasis from renal cell carcinoma (1), duodenal ulcers (2), duodenal Brunner’s gland hamartoma (1), esophageal cancer (1), arterial anomaly of the gastric fundus as a result of pancreaticoduodenectomy (1), gastroduodenal artery bleeding caused by a pancreatic tumor (1).

Chronic anemia or acute anemia with hemodynamic instability requiring transfusions of red blood cells were the most common clinical presentations, as well as hematemesis and melena. Less frequent presentations, such as rectal bleeding and abdominal pain, were also reported.

EUS-guided therapies were performed with sclerosing agents (9: polidocanol in 5 cases, alcohol in 4 cases), cyanoacrylate (15: cyanoacrylate alone in 9 cases, mixed with Lipiodol in 6 cases), coils (3), combined thermal contact therapy and alcohol (1), hyaluronate (3), thrombin (3), band ligation (3), combined band ligation and alcohol (1), combined epinephrine, snare ligature, and polypectomy (1). Procedural details are provided in Table 1.

EUS-guided treatments were performed via an FNA needle in 36 procedures: the size of the needle was 23 G in 3 cases (8.3%), 22 G in 21 cases (58.3%), 19 G in 10 cases (27.8%), and unspecified in 2 cases (5.6%).

### 3.2. Outcome Measures

Before EUS treatment, 25/35 patients (71.4%) underwent endoscopic hemostatic attempts with a range of 1–4 procedures per patient without any success. In 10 cases (28.6%), EUS-guided treatment was performed upfront, mainly because of the difficulty or impossibility, for anatomical reasons, of the angiographic approach.

Doppler ultrasound was used to evaluate the blood flow within target vessels at the end of EUS, obtaining an estimate of the technical result: complete cessation of bleeding was reported in 32/39 procedures (82.0%), a marked decrease of blood flow—in 4 procedures (10.3%), while in 3 cases (7.7%) the result of this technique was not reported. Nevertheless, in 100% of the cases, the bleeding was stopped, and the procedure was considered successful, although 2 patients required 2 consecutive EUS sessions in order to achieve full obliteration of the vessel: one patient with a gastric GIST underwent 2 consecutive sessions of hyaluronate injection, the other patient with a vascular pseudoaneurysm underwent coil insertion followed by thrombin injection. Neither of these 2 patients showed any signs of recurrent bleeding in the time interval [34,36].

The clinical outcome was favorable in 32/35 patients (91.4%), except for three episodes of recurrent bleeding in Dieulafoy’s lesions: in 2 cases, the EUS-guided treatment was repeated successfully, whereas in 1 case the patient was referred for surgical gastric wedge resection [22,32,34]. It is worth mentioning that in the study by Gonzalez and colleagues, another patient experienced rebleeding from a new vascular lesion, which was, however, located in a different site: the new lesion was treated radiologically [32].

No adverse events or complications were reported either during or after the procedures.

The median follow-up was 11.0 months (interquartile range 6–15.5), with only 6 patients followed up for more than 2 years.

## 4. Discussion

In recent years, we have witnessed the progressive expansion of EUS applications, with relevant implications in the diagnostic and therapeutic field. Thanks to its capacity to combine high-quality visualization of anatomical regions that are difficult to explore with other methods, with guidance of transmural access, EUS has become the first choice in the treatment of many conditions [37,38]. EUS offers an unmatched access to abdominal vessels that until now have only been accessible to surgeons and/or interventional radiologists, and it provides a potential alternative approach to patients with refractory GI bleeding who failed treatment with standard hemostatic techniques.

In this setting, EUS may provide a detailed description of the source of bleeding in terms of appearance, size, and anatomical location, but also a unique guide for a direct anatomically vessel-targeted therapy. EUS can guide the placement of coils or delivery of glues in the targeted lesion as well as in the main feeding vessel, and allows the use of hemostatic techniques other than the ones based exclusively on intraluminal endoscopic visualization [39]. Another advantage of EUS-guided procedures is the possibility to make a real-time assessment of the procedural success, as described in almost all of the reviewed cases, by means of Doppler EUS application. Furthermore, a DEP can offer a good opportunity in evaluating the success of the endoscopic treatment. In recent times, the usefulness of EUS has emerged in the context of NVUGIB, especially after the failure of standard endoscopic or radiological therapies, as demonstrated by the case reports and case series revised in our review: in more than 70% of the patients, EUS was performed as a rescue therapy of refractory bleeding.

Glue injection and the use of sclerosing agents were the most commonly used treatments. According to the type of lesion, various techniques were described in the reviewed publications: in pseudoaneurysms and in GISTs, the injection was mainly performed directly inside the lesion or into the feeding vessel, although in one patient with a GIST, hyaluronate was injected adjacent to the bleeding vessel in order to reduce the risk of embolization [34]. Conversely, in most Dieulafoy’s lesions, the tip of the needle was introduced along the course of the bleeding vessel or next to its point of branching, although cyanoacrylates or their combination with Lipiodol can be safely injected into the target vessels. The volume of the injected substance and the treatment modality varied according to the type of lesion, availability of the drug, and the choice of the endoscopist. The complete data are provided in Table 1.

The success rates of the procedures were high, both technically and clinically: EUS-guided treatments allowed the cessation of bleeding in all of the cases, as demonstrated by Doppler signal, with an excellent safety profile, since no adverse events were reported. An excellent clinical outcome was demonstrated by the very low rate of rebleeding during the follow-up with only three episodes of recurrent bleeding.

Nevertheless, although the data on efficacy and safety are promising, the number of cases described in the literature is still too small to obtain solid evidence. All the reported cases come from centers with great expertise and know-how, thus raising questions about the reproducibility of the results. Moreover, the described success rate in case reports and case series suffers from a selection bias: it is very unlikely that unsatisfactory results or failed procedures were taken into consideration for publication by the authors. Quality assessments of the included studies are provided in Table 1. Furthermore, few data are available for follow-up strategies: for pseudoaneurysms, a possible follow-up strategy may be to perform computed tomography angiography at 48 h, at 6 weeks, and then once every 6 months, as suggested by some authors [29,30,31].

EUS-guided angiotherapy is safe and feasible, but for several reasons its use in clinical practice is limited, except for tertiary centers: the main reason is the lack of training in endosonography and the limited availability of EUS in the acute care setting. On the other hand, some technical limitations have to be taken into consideration: iatrogenic bleeding from an extraluminal site (due to the transmural access to deeper tissues) not accessible to standard endoscopy may require urgent salvage angiographic or surgical therapy; thromboembolic events that may also be fatal, especially when injecting glues into vessels; the small caliber of the EUS aspiration channel reduces the possibility to remove blood clots with consequently impaired visualization; luminal contents not aspirated by the endoscope may create artefacts and alter the endosonographic image during the examination [22]; possible immunoreactions when using endovascular therapy with agents such as cyanoacrylate and thrombin [31]; possible induction of infections.

Some of these limitations may be prevented and avoided with a wider use of the accessories and devices specifically designed for EUS [35]. Moreover, the adoption of smaller needles (22 Gauge) may reduce the risk of induced bleeding and the use of less immunoreactive autologous agents, and the correct implementation of antibiotic prophylaxis may help prevent immunoreactions, thromboembolic events, and infections [32,34].

Careful EUS image control during the administration of agents enables to control volume and site of the injection. In order to minimize the risk of embolization and local complications, we should try to standardize the technique of injection, the volume of the injected substance, the size of the coils, and the speed of injection. Another important issue to be aware of is the presence of spontaneous portosystemic shunts and cardiac abnormalities, such as patent foramen ovale prompting right-to-left shunt. Outside the emergency field, prior to endoscopic intervention, preventive angiographic studies may also be proposed.

## 5. Conclusions

EUS-guided therapies are a powerful tool in the management of NVUGIB, especially in cases that are refractory to standard endoscopic treatments, with good efficacy and a good safety profile. The possibility of simultaneously visualizing the lesion, treating it under ultrasonographic guidance, and checking the effectiveness of the treatment are the reasons for which EUS-guided therapies have become so popular in the last few years. Nevertheless, the literature regarding their use in the setting of NVUGIB is scarce, and all data come from case reports and very small series. Larger scale studies, possibly including randomized controlled trials, are needed.

## Figures and Tables

**Table 1 jcm-09-00866-t001:** Literature Review.

Author, Year [ref.]	Type of Article	Included Patients (No.)	Included EUS-Guided Procedures (No.)	Instance of Treatment (First/Repeated)	Type of Lesions (No. of Cases)	Type of Bleeding	Treatment (No. of Cases)	Needle Size	Technical Outcome	Clinical Outcome	Follow-up (Months)	CARE Assessment
Fockens P, 1996 [22]	Case series	3	3	First	Dieulafoy’s lesions (3)	Unspecified GI bleeding	Polidocanol 1% (3)	23 G	Success	Rebleeding after 5 months in 1 patient: surgical gastric wedge resection	14 (median)	19/28
Ribeiro A, 2001 [27]	Case report	1	1	Repeated	Dieulafoy’s lesion (1)	Melena	Thermal contact + alcohol, 2.5 mL (1)	Unspecified	Success	Success	9	15/28
Folvik G, 2001 [28]	Case report	1	1	Repeated	Dieulafoy’s lesion (1)	Severe GI bleeding	Band ligation (1)	n/a	Success	Success	11	12/28
Roach H, 2005 [29]	Case report	1	1	First	Pancreatic pseudoaneurysm (1)	Rectal bleeding and melena	Thrombin 500 IU, 1 mL (1)	22 G	Success	Success	10.5	18/28
Levy MJ, 2008 [21]	Case series	5	5	Repeated	Pancreatic pseudoaneurysm (1), Dieulafoy’s lesion (1), duodenal ulcer (1), GISTs (2)	Overt GI bleeding in 4 cases, transfusion-dependent anemia in 1 case	Alcohol 99%, 4–7 mL (2); cyanoacrylate, 3–5 mL (3)	22 G	Success	Success	12 (median)	23/28
Gonzalez JM, 2009 [30]	Case report	1	1	First	Pancreatic pseudoaneurysm (1)	Arterial leakage during an EUS examination	Lipiodol + n-butyl cyanoacrylate, 2 ampoules (1)	19 G	Success	Success	6	21/28
Lameris R, 2011 [31]	Case report	1	1	First	Visceral pseudoaneurysm (1)	Hematemesis	Thrombin, 7 mL (1)	22 G	Success	Success	10	15/28
Gonzalez JM, 2012 [32]	Case series	6	7	Repeated *	Dieulafoy’s lesions (2), vascular pseudoaneurysms (2), gastroduodenal artery erosion due to a pancreatic tumor (1), arterial anomaly of the gastric fundus (1)	Hematemesis and/or melena	Cyanoacrylate + Lipiodol, 2 mL (5); polidocanol 2%, 4 mL (2)	19 G	Success	Rebleeding after 3 months in 1 patient: EUS-guided treatment	8 (median)	20/28
Kumbhari V, 2014 [33]	Case report	1	1	First	Gastric GIST (1)	Symptomatic anemia	n-butyl 2-cyanoacrylate, 2 mL (1)	19 G	Success	Success	6	11/28
Law R, 2014 [34]	Case series	13	15	Repeated	Esophageal cancer (1), Dieulafoy’s lesions (2), Gastric GISTs (4), marginal ulcer after an RYGB (1), duodenal ulcer (1), duodenal Brunner’s gland hamartoma (1), duodenal metastasis (2), pancreatic pseudoaneurysm (1)	Unspecified GI bleeding	Cyanoacrylate, 2–7 mL (5); alcohol, 0.2–7.5 mL (2); hyaluronate, 3–4 mL (3); coil insertion (1); band ligation (2); band ligation + alcohol, 4 mL (1); epinephrine (1:10,000), 10 mL + snare ligature with polypectomy (1)	22 G	Success§	Rebleeding after 38 months in 1 patient: EUS-guided treatment.	12 (median)	23/28
Jeffers K, 2017 [35]	Case report	1	1	Repeated	Pancreatic pseudoaneurysm (1)	Melena	Coil insertion (1)	22 G	Success	Success	15	10/28
Sharma M, 2019 [36]	Case report	1	2	First	Vascular pseudoaneurysm (1)	Melena	Coil insertion (1); thrombin 4000 IU, 8 mL (1)	19 G	Success	Success	0.5	12/28

Legend: * except for 2 patients; § 1 patient underwent 2 consecutive sessions; G, Gauge; GIST, gastrointestinal stromal tumor; RYGB, Roux-en-Y gastric bypass; EUS, endoscopic ultrasound; GI, gastrointestinal; n/a, not applicable; mL, milliliters; IU, international units.

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
