# Peer review of "Endoscopic Ultrasound-Guided Treatments for Non-Variceal Upper GI Bleeding: A Review of the Literature"

_jcm, 2020, doi:10.3390/jcm9030866_

Round 1
Reviewer 1 Report
As noted by the authors, there are relatively small numbers of cases reported in the literature. While this limits the scope of conclusions that can be drawn, the comprehensive review of published cases at this point in time appears to be a useful contribution to the literature.
Author Response
As highlighted in the discussion and in the conclusion of our paper, the paucity of described cases definitely reduces the strength of evidence for this type of procedure. Nevertheless, in such a rare setting, it is important to try and gather all the data in order to establish (possibly) a starting point for future, more structured studies...and this was one of the main aims of this work.
Thanks for appreciating our work.

Reviewer 2 Report
The purpose of this manuscript is quite vague.
If the authors tried to analyze the effect of EUS-guided treatment for NVUGIB, they should compare the EUS-guided group vs. group without EUS guidance. Literature review for cases and retrospective studies with only a few patients cannot lead the readers to the right way. I would recommend the authors to perform a meta-anlysis.
Author Response
We truly understand your concerns. As highlighted by another reviewer in the first round, the main issue of this topic is the lack of structured data in literature: thus, we extensively revised the first version our paper, in order to further analyze the available evidence and to resume it (as shown in table 1) in order to make it well-structured and possibly more usable...this was our purpose. The CARE checklist was also used to stratify the strength of the single papers, since tools such as ROBIN-I for bias assessment are not applicable for case reports and case series.
We never meant to make a comparison between the EUS-guided technique and other techniques, simply because the amount of cases and patients is too scarce (even compared to the tens of thousands of other techniques) and the approach by the different authors is too heterogeneous to make a direct comparison with a control group....moreover, the difference between the single type of bleeding lesions (pseudoaneurysms, Dieulafoy lesions, tumors...) would require a possible plethora of various techniques (i.e. radiological, surgical, “standard” endoscopic”...) which would make a standard comparison even more inhomogeneous. For the same reason, we believe that modifying the structure of the paper into a meta-analysis couldn't be of any help: since there are no data coming from original studies, case control series or RCTs, the strength of the derived evidence would remain extremely scarce. On the other hand, we extensively reviewed all the available literature following the journal invitation for a review.
Once again, we highlight that this paper is not meant to give any direct recommendation related to the EUS-guided technique, as pointed out several times in the text: the evidence in literature is not enough to do this.

Round 2
Reviewer 2 Report
Thank you for meticulous revision.
This manuscript is a resubmission of an earlier submission. The following is a list of the peer review reports and author responses from that submission.
Round 1
Reviewer 1 Report
Comprehensive compilation of case series and case reports regarding non-variceal upper gastrointestinal bleeding treated with eus guided therapy. 33 patients are reported. The largest single prior case series included 13 patients. Table 1 summarizing the literature was nicely presented. While reading this I found myself interested in an expanded discussion of ongoing issues, including: How often was the bleeding vessel directly injected/treated versus a feeding vessel or injection into tissue next to the bleeding vessel? What range of volume of injectate was injected for each substance? What factors should be considered prior to eus guided therapy to ensure that the risk of embolization is not increased or is acceptable? For pseudo-aneurysms, what type of follow up imaging would be advised to ensure adequate response to eus guided therapy?
Reviewer 2 Report
In their article „Endoscopic ultrasound-guided treatments for…” the authors reviewed the literature on EUS for the treatment of non-variceal upper GI bleeding. They reviewed the results of the procedure in 33 patients that were published in 7 case reports and 4 case series. The cause of GI-bleeding were Dieulafoy’s lesions, pancreatic pseudoaneurysms and GIST. The treatment modalities did vary from patient. Most patients were treated with injection of cyanoacrylate, others underwent sclerotherapy, injection with fibrin glue, coil embolization or band ligation. The reported technical outcome was successful in all patients.
The study is of interest to gastroenterologist as it shows that EUS guided therapy might be of use in patients who are refractory to other treatment modalities. Nevertheless, there is a heavy publication bias, since it is very likely that only case reports and case series were published in which successful treatment was achieved. This, and the lack of controlled studies, should be discussed in more detail. Accordingly, the title should be for example: “Endoscopic ultrasound-guided treatments for non-variceal upper GI-bleeding: a review of the literature”.
Reviewer 3 Report
This study looks like a systematic review and meta analysis. However, due to limited studies and number of patients, the authors did not perform a systematic review.
Major concerns;
Please add quality assessment of each studies using tools such as ROBANs.